# Structural Transformation and Creativity Induced by Biological Agents during Fermentation of Edible Nuts from *Terminalia catappa*

**DOI:** 10.3390/molecules26195874

**Published:** 2021-09-28

**Authors:** Oluwatofunmi E. Odutayo, Bose E. Adegboye, Emmanuel A. Omonigbehin, Tolulope D. Olawole, Olubanke O. Ogunlana, Israel S. Afolabi

**Affiliations:** 1Biochemistry Department, College of Science and Technology, Covenant University, Ota 100122, Nigeria; oluwatofunmi.obaseki@gmail.com (O.E.O.); bose.adegboye@covenantuniversity.edu.ng (B.E.A.); tolulope.olawole@covenantuniversity.edu.ng (T.D.O.); olubanke.ogunlana@covenantuniversity.edu.ng (O.O.O.); 2Molecular Biology Laboratory, College of Science and Technology, Covenant University, Canaanland, Ota 100122, Nigeria; adedayo.omonigbehin@covenantuniversity.edu.ng

**Keywords:** *T. catappa*, seed, nuts, food, processing, fermentation, phytochemical

## Abstract

*Terminalia catappa* L. (tropical almond) is a nutritious fruit found mainly in the tropics. This study is aimed to establish the naturally biotransformed molecules and identify the probiotic agents facilitating the fermentation. The aqueous extracts from both the unfermented and fermented *T. catappa* nuts were subjected to gas chromatography/mass spectrometry (GC/MS) analysis. Syringol (6.03%), glutamine (1.71%), methyl laurate (1.79%), methyl palmitate (1.53%), palmitic acid (5.20%), palmitoleic acid (2.80%), and methyl oleate (2.97%) were detected in the unfermented nuts of the *T. catappa*. Additionally, two of these natural compounds (palmitic acid (4.19%) and palmitoleic acid (1.48%)) survived the fermentation process to emerge in the fermented seeds. The other natural compounds were biotransformed into 2,3-butanediol (1.81%), butyric acid (16.20%), propane-1,3-diol (19.66%), neoheptanol (2.89%), 2-piperidinone (6.63%), palmitoleic acid (1.18%), formamide, n-(p-hydroxyphenethyl)- (2.80%), and cis-vaccenic acid (1.69%) that newly emerged in the fermented seeds. The phytochemical compounds are likely carbon sources for the organisms facilitating the biotransformed molecules and product production. Four (4) potential probiotic bacteria strains, namely, Probt B1a, Probt B2a, Probt B4a, and Probt B4b, were isolated from the fermented nut. *Enterococcus faecum*, and *Enterococcus faecalis* were the organisms identified as driving the fermentation of the seeds. All strains were gram-positive, catalase-negative, and non-hemolytic, which suggests their harmless nature. N-(p-hydroxyphenethyl)-) was associated with fermentation for the first time, and neoheptanol was discovered as the main alcoholic molecule formed during the fermentation of the seeds. This fermentation is a handy tool for bio-transforming compounds in raw food sources into compounds with nutritious and therapeutic potentials.

## 1. Introduction

Plant seeds supply food for animals and humans due to their rich nutrients [1]. The seeds of sunflower, soybean, peanut, cotton, and palm nut were food sources for ages. The seeds from many edible fruits (apple, orange, watermelon, lemon, avocado pear, and mango) are still under-utilised [1]. The unpleasant taste and the excessive anti-nutrient content of these under-utilised seeds remain a major source of concern [1]. Studies on the phytochemical, bioactive, and nutritional components of some neglected seeds have led to the realisation that they can alternatively nourish consumers with health-promoting substances such as secondary plant metabolites micronutrients and vitamins [2,3,4]. Fermentation, one of the ancient food processing methods, has been widely employed in improving taste, flavour, and reduction of anti-nutrients in food substances to transform the neglected, under-utilised, or non-utilised seeds to a useful state [5,6]. Metabolic processes that occur during fermentation produce a chemical transformation of organic substrates with the aid of enzymes secreted by the fermenting microorganisms. Consequentially, fermented foods are products consisting of modified compounds that are different from their original nature. In a recent study by Odutayo, et al. [7], the Gas Chromatography/Mass Spectrometry (GC/MS) analysis of the natural and fermented under-utilised kernel of *Chrysophyllum albidum* (African star apple) gives a vivid illustration of the biotransformation of the compounds in the raw seeds to that in the fermented seeds. Fermentation of foods results in products with diverse microbial communities and the bacteria in these communities can be similar to probiotics [8].

*Terminalia catappa* is also known as tropical almond belonging to the family Combretaceae. It is one of the under-utilised tree species found in the tropics [9]. They are planted mainly for shade and ornamental purposes in the tropics and are mostly found in parks and home gardens. The fruit has an edible fibrous pulp usually eaten as forage snacks by children [10]. It contains a single seed rich in protein, which could serve as an alternative source of nourishment to reduce malnutrition in impoverished children [10]. This fruit has a husk, a fibrous and porous pericarp, an exocarp relatively thin and smooth, with a stony endocarp enclosing the edible nut [11]. Nuts from *T. catappa* fruits are small in size and difficult to extract, and these are likely factors that have contributed to its under-utilisation. The sun-dried nut has about 38–54% yield of light yellow oil that is edible; it, however, becomes turbid on standing [11]. *T. catappa* seed is also rich in vitamin E and high in unsaturated fat [11]. In some parts of South America, the oil is used in cooking, while the nuts are freshly consumed immediately after extraction from the shell. Additionally, it is well preserved by smoking and consumed for even up to a year later [11]. According to the American Standard for Testing Material (ASTM) methods, the use of the oil from this seed as an affordable biodiesel source was tested affirmatively according to the American Standard for Testing Material (ASTM) methods [12].

Fermentation is an ancient tool engaged by humans to process foods [3]. It has a preference for enhancing the quality of nutrients and taste of foods, as well as improving their shelf-life, safety, and digestibility [13,14,15]. Fermentation is often driven by the activities of microorganisms and is increasingly used in the field of bioengineering and biotechnology to achieve desirable food products [14,16,17]. Natural, solid-state, and liquid fermentations are the three major processes of fermentation. These fermentation processes can further be categorised by the biochemical principle of action propelling it into an alcoholic, lactic acid, acetic acid, or butyric fermentation [18,19,20]. The natural submerged or spontaneous fermentation is the most preferred method for processing maize, millet, and sorghum into pap in homes [21]. Therefore, a study of this spontaneous fermentation should provide the best opportunity to understand the diverse organisms and phytochemicals that may be eventually consumed domestically [7,22]. The lactic acid bacteria is a major fermenting microorganism during the natural (spontaneous) fermentation of foods [16]. This awareness encourages their isolation as probiotic bacteria and their commercial availability and is therefore utilized for several decades in the food industry [23].

A report on using the fermentation technique to process the seeds of *T. catappa* and the associated biotransformation mechanisms does not exist. In this study, we engage gas chromatography-mass spectrometry (GC-MS) analysis to establish the phyto-compounds that may be obtained from the fermented kernel of *T. catappa* and its raw form so to appreciate biotransformation process during the fermentation. An attempt was also made to identify and separate possible pure strains of probiotic lactic acid bacteria facilitating the fermentation.

## 2. Results

### 2.1. Effects of the Fermentation on pH of the Extracts

The fermentation led to the reduction in pH of the extracts from 7.8 to 7.5 (Table 1).

### 2.2. Isolated Organisms Encouraging the Fermentation

The four isolates of lactic acid bacteria (LAB) sourced from the previously fermented *T. catappa* nuts were as follows: ProbtB1a, ProbtB1b, ProbtB4a, and ProbtB4b. All the isolates were gram-positive, non-haemolytic, and catalase-negative. Probt(B1a, B1b, and B4b) were tiny, circular, milky, and coccobacilli shaped when viewed under the microscope as depicted from the outcome of the morphological identification performed. Probt B4a was also tiny, circular, milky, and rod-shaped when viewed under the microscope. The Analytical Profile Index (API) system was used in obtaining the names for the isolates. This system was limited in identifying the isolate ProbtB4a, but facilitated the identification of Probt (B1a, B1b, and B4b) as *Lactobacillus pentosus*, *Lactobacillus casei*, and *Lactobacillus plantarum*, respectively. These organisms were isolated from the nuts of *T. catappa* for the first time in this study. Figure 1 indicates the gel image obtained during the genomic characterization of the organisms. The outcome genomic analysis of these identified organisms revealed the organisms as different types of LAB. The sequences as submitted to the NCBI database, and their respective accession numbers are as indicated (Table 2).

Bands show the 16S rRNA characterisation for the ten (10) isolates. Keys: L—DNA ladder (100 bp), 1—ProbtB4b, 2—ProbtB3, 3—ProbtB4a, 4—ProbtB1a, 5—ProbtB1b, 6—ProbtB2b, 7—ProbtB2a, LAB-Lactic Acid Bacteria. Note: Bands, not labelled, are for isolates from another seed.

#### Genomic Analysis of Identified ORGANISMS

The four isolates of lactic acid bacteria (LAB) sourced from the previously fermented *T. catappa* nuts were as follows: Probt B1a, ProbtB1b, ProbtB4a, and ProbtB4b. All the isolates were gram-positive, non-haemolytic, and catalase-negative. Probt(B1a, B1b, and B4b) were tiny, circular, milky, and coccobacilli shaped when viewed under the microscope as depicted from the outcome of the morphological identification performed. Probt B4a was also tiny, circular, milky, and rod-shaped when viewed under the microscope. The Analytical Profile Index (API) system was used in obtaining the names for the isolates. This system was limited in identifying the isolate ProbtB4a, but facilitated the identification of Probt(B1a, B1b, and B4b) as *Lactobacillus pentosus*, *Lactobacillus casei*, and *Lactobacillus plantarum*, respectively. These organisms were isolated from the nuts of *T. catappa* for the first time in this study.

### 2.3. Isolated Organisms Facilitated Transformation of Molecules during the Fermentation

This study is the first to provide the detailed phytochemical compounds present in the freeze-dried extracts from the unfermented and the naturally fermented *T. catappa* nuts (Table 3 and Table 4). Syringol (6.03%), glutamine (1.71%), Methyl laurate (1.79%), Methyl palmitate (1.53%), palmitic acid (5.20%), palmitoleic acid (2.80%), and methyl oleate (2.97%) were the seven (7) major compounds detected in the unfermented extract. From this study, it could be inferred that these seven compounds (Table 3) contributed to supporting growth of the probiotics facilitating the fermentation process [24]. Ten (10) major compounds, namely 2,3-butanediol (1.81%), butyric acid (16.20%), Propane-1,3-diol (19.66%), Neoheptanol (2.89%), 2-piperidinone (6.63%), Palmitoleic acid (1.18%), palmitic acid (4.19%), palmitoleic acid (1.48%), Formamide, N-(p-hydroxyphenethyl)- (2.80%), and cis-vaccenic acid (1.69%), were identified as the major metabolites in the fermented extract (Table 4). These newly formed metabolites could also serve as a simple explanation for fermentation as a biotechnological tool for the formation of new products due to the activities of microorganisms taking part in the fermentation process.

## 3. Discussion

### 3.1. Identified Organisms Encouraging the Fermentation

Catalase-negative, gram-positive, and negative haemolytic reactions are basic ways to identify lactic acid bacteria (LAB) that should be regarded as safe. LAB is the most common species used as probiotics in the food industry, and they have been generally regarded as safe (GRAS). The API system has been used to identify bacteria over time as an advancement over the manual preparation of sugars for the biochemical characterisation of isolates in the laboratory. As the end product of their fermentation, LAB produces lactic acid. Thus, the API system for identifying LAB operates the principle of the acidification of the sugar with the lactic acid released by LAB, thereby showing their ability to utilise the sugars (Table 5).

The most commonly used probiotics in foods have been the lactic acid bacteria (LAB). Non-hemolytic, gram-positive and catalase-negative reactions are preliminary ways to identify potential probiotic LAB strains. Traditionally, most LAB were identified based on their morphological, physiological, and carbohydrate fermentation patterns [25]. However, the 16sRNA characterization is considered a more acceptable technique for identifying bacteria isolates to sub-species level [26]. The 16sRNA sequencing revealed that all the isolates in this study belong to the Enterococcus species. Enterococci are gram-positive cocci occurring in pairs or short chains, non-spore-forming, catalase and oxidase-negative, and facultative anaerobic [27,28]. The *Enterococcus* genus belongs to the lactic acid-producing bacteria group. It represents the third-largest LAB genus after *Lactobacillus* and *Streptococcus*. It contains 37 species that have been classified based on phylogenetic assessment with the use of 16sRNA sequencing and DNA-DNA hybridisation [25]. Enterococci are ubiquitous microorganisms predominant in places such as water, plant, soil, foods, and the gastrointestinal tract of humans and animals [29]. They have served as probiotics and starter culture in food fermentation due to their biotechnological traits, like enzymatic and proteolytic activities or protective cultures in food bio-preservation due to their produced antimicrobial bacteriocins often known as enterocins [29]. A recent debate is generated on the use of enterococci in foods or as probiotics due to their opportunistic pathogenicity [29]. Therefore, strict monitoring of any new enterococcal probiotics is recommended to ensure these organisms’ safety and harmless nature before applications. The non-hemolytic reaction expressed by the isolated enterococcal strains in this study ascertain their safety. In addition, these isolated strains, namely *E. faecalis*, *E. durans*, and *E. faecum* exists commonly in cheeses made from raw or pasteurised milk.

### 3.2. Isolated Organisms Enhanced Transformation during the Fermentation

Fermentation is an anaerobic process converting glucose to ethanol and carbon dioxide. Organic acids and alcohols are usually produced during fermentation [30]. Lactic acid was not the major molecule in the fermentation of the seeds from *T. catappa*, as earlier reported for the seeds obtained from *C. albidum* [7]. Butyric acid (P4) and cis-vaccenic acid (P7) were the predominant acids produced during the fermentation of the seeds of *T. catappa* instead of lactic acid (Figure 2).

The many compounds identified in this study are consistent with the findings obtained in other similar plant materials [31]. Plants, such as the seeds of *T. catappa* used in this study, are complex in structure with myriads of molecules that may be of health benefits [32]. Butyric acid is peculiar to the butyric acid fermentation type previously identified with glucose and xylose metabolism by *Clostridium tyrobutyricum* [33]. The butyric acid may be a product resulting from the breakdown of the dietary fibres in the *T. catappa* nut by the probiotics that enhanced the fermentation process. Butyric acid is an aliphatic organic acid synthesised by the action of microbiota in the human gut that is often associated with anaerobic conditions. The microbiota degrades dietary fibres, which are enhanced by the anaerobic environment in the gut [34]. Butyric acids, propionic acids, and acetic acids are the primary short-chain fatty acids among the products of such catabolism. Humans do not have the enzymes needed for the metabolism of the dietary fibres [34,35]. These major SCFAs have the possibility of playing an essential role in preventing and managing diseases such as certain types of cancer, bowel disorders, and metabolic syndrome [35].

Cis-vaccenic acid (P7) is a bio-active molecule implicated with milk, wines, and vinegar. It may exist as a natural compound of the leaf of *Crateva adansonii* DC [36]. The compound was previously linked as a derivative of oleic acid. The proposed mechanism for forming the cis-vaccenic acid (P7) from methyl oleate in this study (Figure 2) further substantiated P7 as a derivative of oleic acid [37,38,39]. Cis-vaccenic acid is an Omega-7 fatty acid and a stereoisomer of vaccenic acid found in the oil of *Hippophae rhamnoides* that is commonly known as sea buckthorn [40,41]. Diets rich in omega-7 fatty acids often increase HDL cholesterol levels and lowering LDL cholesterol levels which explains their beneficial health effects [42]. The cis-vaccenic acid plays a role in reducing unsaturated fatty acid toxicity by yeast *Saccharomyces cerevisiae*. The reduction of the palmitoleic acid content in the cells is consistent with cis-vaccenic acid formation. *S. cerevisiae* transforms palmitoleic acid into cis-vaccenic acid to survive the absence of triacylglycerol [43].

Neoheptanol (P6), a recently discovered alcoholic molecule, was mainly formed during the fermentation of the seeds of *T. catappa* instead of the ethanol reported for the fermentation of the seeds obtained from *C. albidum* [7]. Fermentation of the seeds from *T. catappa* also produced other alcoholic products, such as butane-2,3-diol (P3) and 2,2-dimethylpentan-1-ol in addition to neoheptanol (Figure 1). These further established the fermentation used to process the seeds from *T. catappa* in this study. The neoheptanol (P6; 2,2-dimethylpentan-1-ol) is a rare compound implicated mainly with vanilla, a fermentation product [44,45]. The neoheptanol may result in a metabolic product from pathways that either synthesis the aromatic amino acid or the aldehydes [46]. The alcoholic compound (2,2-dimethylpentan-1-ol) was presently involved with another fermentation product (*T. catappa*) in this study (Figure 1). 

A yet-to-be-reported compound (P1; Formamide, N-(p-hydroxyphenethyl)-) was first associated with fermentation in this study (Figure 1). Although, tyramine and β-phenylethylamine, which are derivative compounds, were also recently implicated with fermented products. The tyramine formation was associated with the bioactivity of *Lactobacillus plantarum* detected in this study [24,47,48,49]. Tyramine, N-formyl-, belongs to the class of substituted amides. The bonds in substituted amides are quite stable and found in the repeating units of protein molecules (peptide linkage). The synthesis of amides from both alcohols and amines produces hydrogen. This reaction step is facilitated by the dehydrogenation of hemiaminal intermediates, and subsequently an aldehyde reaction with the amine [50]. The metabolite, tyramine, N-formyl could thus be a major antimicrobial compound with antimicrobial activities in the fermenting probiotics [24]. 

2-Piperidinone (P2) detected in this study was similarly linked to fermented products [51]. The detection of 2-piperidinone as one of the major compounds in fermented *T. catappa* nuts is notable. 2-piperidinone is highly sought after by chemists because it serves as an intermediate in preparing other chemicals. It is a vital commodity in the synthesis of nylon polymers [52]. Many efforts have also been put into producing this compound biologically using *Escherichia coli* [30,34,53]. The breakdown of the sugars (Table 5) inherent in this nut may have yielded 2,3-butanediol, a plant growth-promoting compound that results from glucose fermentation [46]. The detection of 2,3-butanediol after fermentation is similar to the findings in fermented horse gram sprouts [46]. The production of the butane derivatives occurs during the fermentation of *T. catappa*, which is similar to the findings on the fermented *C. albidum*. Three molecules with possible root from butane metabolism, methyl butanoate (I-3), butyric acid (P4), and butane-2,3-diol (P3), was implicated during fermentation of the seed *T. catappa* (Figure 2), instead of the 1,3-dihydroxybutan-2-one and (2E)-2,3-dihydroxybut-2-enal that was reported for the fermentation *C. albidum* [7,36]. The fermentation of *T. catappa* was also linked to propane derivatives (P5; propane-1,3-diol) as an intermediary compound (Figure 1). Although, two compounds ((1E)-prop-1-ene-1,2-diol and 2-hydroxypropanal) were the derivatives of propane derived from the fermentation of the seeds of *C. albidum* [7]. 

Methyl butanoate (I-3) had been linked with the fermentation process [36]. The numerous reports linking octane (I-1) to fermentation were mainly on the biofuel quality of fermented products, as revealed by their octane rating [54,55]. The involvement of octane (I-1) metabolism with the fermentation process is yet to be reported as proposed in this study (Figure 1). Neither has the octane (I-1) reaction with methyl butanoate (I-3) elucidated during fermentation as established in this study (Figure 1). Dodecane (I-2) was previously implicated in the process of fermentation. It is a vector of oxygen in the biosynthesis of fumarase during fermentation [56]. Understanding such metabolism has led to the advancement of biosynthesis of vitamin A [57]. Palmitic acid and palmitoleic acid exist in both the unfermented and fermented extracts. The survival of these two fatty acids indicates that their functional groups are gradually modified by the slight acidic changes transforming the phytochemicals during this fermentation [7]. The two compounds were not completely metabolised like other compounds completely eliminated after the fermentation indicates a gradual modification process. It may require a more extended period of fermentation to be completely metabolised. The slight change in acidity (Table 1) during the fermentation of this nut may be responsible for the slow modification of the acidic functional groups of both compounds. An extended period of fermentation beyond the 72 h used in this study may completely metabolise these compounds (palmitic acid and palmitoleic acid). This fermentation of *T. catappa* produces a pH change of 3.0, similar to the slight pH change earlier reported during fermentation [58]. The slight difference in acidity was sufficient to induce a change in the activity of microorganisms that drives the modification of compounds [58,59]. A pH change from 6.32 to 4.25 (a margin of 2.07) produced conspicuous biochemical changes during fermentation [60,61]. Other similar fermentation studies with pH change from 5.0 to 7.0 (a margin of 2.0) resulting in a chemical modification depend on the type of compounds present during the fermentation. The pH change did not affect the amino acids but immensely impacted the carbohydrate structures [62]. Such slight change in acidity led to the modification of polyphenols, γ-aminobutyric acid, citric acid, and other compounds during microbial fermentation or in other living systems [63,64,65,66,67,68,69]. These findings also signify that the *T. catappa* nut is a rich source of dietary fatty acid in raw and fermented form.

### 3.3. Benefits of the Major Compounds Present in the Unfermented Freeze-Dried Extract of T. Catappa Nuts

Among the seven major compounds in the unfermented extract, syringol had the highest area percentage (6.03%). Being a flavour compound, it shows its major contributory role in the natural aroma of this nut [70]. This naturally occurring aromatic organic compound is the main chemical responsible for the smoky flavour in food preparation by smoking [71]. Syringol is very abundant in hardwoods as those from oak and walnut [72]. Glutamine is a non-essential and essential amino acid that often facilitates the synthesis of proteins and glutathione synthesis, a metabolic process that produces energy. The status of antioxidants and immune function are also optimally maintained or regulated by glutamine. It regulates numerous gene expressions as well [73]. Glutamine is synthesized normally in the body in adequate amounts. Still, this amino acid could become conditionally essential during periods of metabolic stress. Hence, because of its numerous metabolic functions, a glutamine-rich diet or supplements is recommended for patients when the need arises [73]. Fatty acids are important dietary sources of fuel for animals [74]. They are also important structural components for cells. The methyl laurate, methyl palmitate, palmitic acid, palmitoleic acid, and methyl oleate are the fatty acids and fatty acid esters identified among the seven major compounds in the unfermented extract. This explains the rich oil content earlier reported by Ladele, et al. [75] for this nut.

## 4. Materials and Methods

### 4.1. Procedure for Collecting and Identifying Plant

The fruits from *T. catappa* tree were picked from the environment of the University of Ibadan, Oyo State, which is located on 7.4443° N, 3.8995° E geographical axis of the world. The *T. catappa* fruits (Voucher number: FHI 112775) were identified and authenticated by experts from the Forest Research Institute of Nigeria (FRIN), Ibadan, Nigeria. Ethical permission (CHREC/49/2020) was secured from the Covenant Health Research Ethics Committee, Canaanland, Ota, Nigeria, before the commencement of this study.

### 4.2. Preparation of Flour from Kernel

The *T. catappa* fruits manually were cracked to collect the nuts after it was sun-dried. The dried nuts collected from the kernels of the seeds were immediately processed into a fine powder using an electronic homogeniser. The derived flour was divided into two portions, with one part to be fermented. The other part of the flour from the raw-unfermented seeds served as the control for this experimentation.

### 4.3. Production of Both the Unfermented and the Fermented Freeze-Dried Extracts

The unfermented and fermented aqueous extracts were prepared according to the method described by Odutayo, Omonigbehin, Olawole, Ogunlana and Afolabi [7]. The processed ground fine flour from the nuts was steeped and stirred in distilled water (1:3 *w*/*v*), and further given time to settle for 24 h, under room temperature (26–33 °C). The homogenate was passed through filter paper (Whatman no. 1) to obtain a filtrate, which was separated into two portions (A and B). These two filtrates were later used for the isolation of the probiotic organisms. One portion of filtrate (A) was immediately freeze-dried (−20 °C) to prevent the initiation of fermentation. The second portion of the filtrate (B) was kept in an airtight container to normally ferment anaerobically for 72 h under ambient temperature (26–33 °C). This filtrate B was immediately freeze-dried (−20 °C) using Telstar Cryodos-50 freeze dryer to also prevent further fermentation after the 72 h. The dried extracts obtained from filtrate A and filtrate B were afterwards labelled as the unfermented (C) and fermented (D) freeze-dried extract, respectively (Figure 3). The pH of the fermentation meshes before and after fermentation was determined using a pH meter (pHep H196107). This instrument was pre-standardised with solutions of known pH (4, 7, and 9) before usage.

### 4.4. Characterisation of the Isolated Organisms Enhancing the Fermentation

#### 4.4.1. Isolation of Probiotic Bacteria from the Fermented *T. catappa* Seeds

This was done according to the method described by Odutayo, Omonigbehin, Olawole, Ogunlana and Afolabi [7]. The homogenate from the fermented *T. catappa* seeds was dispensed into a sterile de Man Rogosa Sharpe (MRS) agar (Oxoid medium) plate previously prepared as instructed by the manufacturer. This selective media (de Man Rogosa Sharpe, MRS) agar that encourages the growth of LAB, but excludes the growth of other microbes, was used to isolate the organisms. A sterile nichrome inoculating loop was used for this purpose. The prepared sample was followed by streaking to assist in observing the growth of conspicuous bacteria colonies. The plates were maintained at a very low level of oxygen using microaerophilic gas packs (Thermo Scientific Ltd., Basingstoke, UK) at 37 °C for 48 h. Single colonies were repeatedly isolated after each incubation and subjected to the process of streaking on the sterile MRS agar plates. The repeated processes described were mainly discontinued when pure cultures of the isolates were derived.

#### 4.4.2. Safety Evaluation, Gram and Catalase Reaction of Isolates

Safety evaluation of isolates was carried out as described by Padmavathi, Bhargavi, Priyanka, Niranjan and Pavitra [23]. Blood agar was prepared by aseptically collecting and adding 5 mL of healthy human blood into 100 mL of sterile Mueller-Hinton. The blood and agar mix was aseptically and gently shaken to obtain a homogenous blood agar. Sterile blood agar plates were afterward prepared by aseptically pouring the blood agar into sterile Petri dishes and left to gel. Spot inoculation of the new 24 h-old culture isolated from the previously generated culture was done on the sterile blood agar plates. The inoculated plates were microaerophilically incubated at 37 °C for 48 h. After incubation, the plates were viewed carefully, and isolates showing zones of haemolysis were not considered safe for further study. Isolates were also sub-cultured on sterile MRS agar plates and maintained at a very low level of oxygen at 37 °C for 24 h. The recently-made cultures were evenly spread on the surface of slides and Gram-stained to examine their shape and nature of Gram-reaction critically. The reaction of the isolates with hydrogen peroxide (H_2_O_2_) was facilitated to test for their catalase activity. An aliquot of the hydrogen peroxide (H_2_O_2_) solution (20 µL) was dispensed on each of the isolates that were previously mounted on a slide. The formation of bubbles was used to detect a positive catalase reaction, and a catalase-negative reaction was confirmed when otherwise observed.

#### 4.4.3. Biochemical Properties of the Isolated LAB

This was done according to the method described by Odutayo, Omonigbehin, Olawole, Ogunlana and Afolabi [7]. The pattern of sugar metabolism expressed by the isolates was identified with the Analytical Profile Index (API) 50 CHL Kit (Biomereux Ltd., Marcy-l’Etoile, France) using the procedure provided by the manufacturer. The LAB isolates were freshly mixed with the API 50CHL basal medium after they had been sub-cultured 24 h. the mixture was then dispensed into the wells on the API 50 CHL test strips, incubated at 37 °C for 48 h before visual examination. The manifestation of a colour change, from purple (scale 1) to green (scale 3) through to yellow (scale 5), served to indicate a positive (value > 3) or negative outcome at the end of 48 h experimentation. The results were after that compared with those presented in the API^®^ databases using the APIwebTM. The same databases were employed to generate the names of the isolates with the aid of the outcome of their carbohydrate utilisation pattern.

#### 4.4.4. Genomic Characterisation of the Isolates

DNA was isolated from the freshly sub-cultured 24-h old LAB isolates using Quick-DNA™ Miniprep Plus Kit by Zymo Research (Inquaba Biotechnology, Pretoria, South Africa). The procedure for isolation was carried out according to the manufacturer’s instructions. 16s forward Primer: 5′-AGGTTGTCTGCTCA-3′ and 16s reverse Primer: 5′-TCGGTCCTTGTCGACT-3′ (Inquaba Biotechnology, South Africa) was used in the bidirectional sequencing. A 100 bp DNA ladder was used as the molecular marker. Polymerase chain reaction (PCR) was carried out in total volumes of 25 µL containing 50 ng genomic DNA, 20 mM primers, 1X Taq buffer, 0.2 mM deoxyribonucleotide phosphate (dntp), 1.25 units Taq polymerase, and DNA and RNAse free water. PCR products were separated on an agarose gel pre-stained with ethidium bromide. Amplicons of 1500 bp obtained were sent to Inquaba biotechnology South Africa sequencing service lab to obtain the sequences of the isolates.

##### Sequencing of the 16sRNA Amplicons

PCR products were cleaned using ExoSAP-ITTM PCR product clean-up reagent. Exo/SAP master mix was prepared by adding 50 µL of exonuclease I (20 U/µL) and 200 µL of shrimp alkaline phosphatase (1 U/µL) into a 0.6 mL microcentrifuge tube. 2.5 µL of the Exo/SAP mix was added to 10 µL of each amplified PCR product, appropriately mixed, and incubated at 37 °C for 30 min. The reaction was stopped by heating the mixture at 95 °C for 5 min. Sequencing was thereafter carried out using the Applied Biosystems™ BigDye™ Terminator v3.1 cycle sequencing kit according to the instructions provided by the manufacturer. The labelled products were cleansed using the ZR-96 DNA sequencing clean-up kit according to manufacturer’s instruction. The cleaned products were injected on the Applied Biosystems ABI 3500XL Genetic Analyser with a 50 cm array, using POP7.

##### Computational Analysis of the Genomic Sequences

Sequences were assembled using Geneious prime (version 2021.0.1). Blasting was done for the assembled sequences using the Basic Local Alignment Search Tool for nucleotide sequences from the National Centre for Biotechnology Information (NCBI).

### 4.5. Sample Preparation for Gas Chromatography-Mass Spectrometry (GC-MS) Analysis

The remaining freeze-dried extracts (5 g) from the unfermented (C) and the fermented (D) filtrates were dissolved again in 10 mL of ethanol solution to obtain the aqueous extracts that was used for the GC-MS analysis.

#### Procedure for GC-MS Analysis

The procedure described by Odutayo, Omonigbehin, Olawole, Ogunlana, and Afolabi [7] was followed in this experimentation. The Shimadzu GC-MS (Model-QP2010SE, Osaka, Japan) used for this experimentation was equipped with a mass spectrometry detector that was set at an equilibrium time of 1.0 min, with an ion source and interface temperatures of 230 and 250 °C, respectively. The detector was set to gain 1.28 kV + 0.00 kV and a threshold of 2200 while the operation is initiated.

An aliquot of the aqueous extracts (1.0 µL) was introduced at 250 °C into the inlet of the GC-MS attached a pre-conditioned Optima 5 MS capillary column (30 × 0.25 mm) using a splitless mode. The capillary column possessed a film thickness of 0.25 µL, at 60 °C. The sampling time (2.00 min), and a linear velocity flow control mode were adopted. A purge flow (3.0 mL/min), a split ratio (5.1), and a pressure (144.4 kPa) was adopted to generate a total flow of 22.6 mL/min, and a column flow of 3.22 mL/min at a linear velocity (46.3 cm/s). The oven temperature was set to perform increasingly at 60, 120, and 290 °C after operating for a period of 2, 2, and 3 min, respectively. The separated compounds (Appendix A and Appendix A) were tentatively identified using the NIST Mass Spectral Search Program for the NIST/EPA/NIH Mass Spectral Library (version 2.0 g, National Institute of Standards and Technology, Gaithersburg, Maryland, USA). The databases of Human metabolome, PubChem, or yeast metabolome were employed to determine their common or scientific names following the identification of the compounds. The major detected compounds with the similarity index of (≥85%) and/ or % peak area that was >1% were considered for this study. These inclusion criteria from the chromatographs were manually implemented since the GC-MS was in scan mode during the analysis. The structural drawing of the compounds and the proposed bio-transformational mechanisms were performed with ChemAxon MarvinSketch software (15.9.14, ChemAxon limited, Budapest, Hungary).

## 5. Conclusions

*L. pentosus*, *L. casei*, and *L. plantarum* were the organisms identified as playing a critical role in driving the fermentation of the edible seeds of *T. catappa*. These organisms were considered safe for consumption, as illustrated in this study. Their application is therefore recommended for possible use as functional food products. However, the isolated LAB strains can be characterised further for more probiotic properties in vitro, in vivo, and molecularly to ascertain their use as probiotics in the food and pharmaceutical industries. N-(p-hydroxyphenethyl)- was linked to fermentation for the first time, and neoheptanol was discovered as the primary alcoholic molecule formed during the fermentation. The biochemical mechanism action for fermentation of the seeds of *T. catappa* followed that of butyric acid fermentation, since the butyric acid was produced during the process. The biotransformation during the fermentation of the seeds, the involvement of octane, and the possible intermediary reactions supporting the high octane ratings of fermented products was postulated for the first time in this study. The illustrated biochemical mechanisms should help to biosynthesize these compounds for use as a starter culture to nurture these organisms in the future.

## Figures and Tables

**Figure 1 molecules-26-05874-f001:**
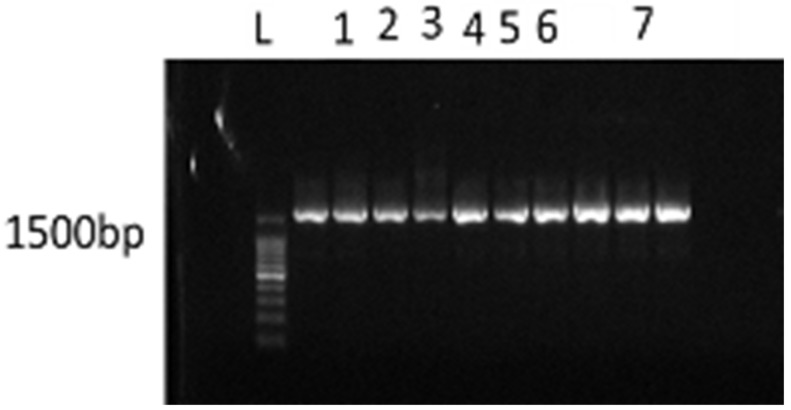
The characterisation of the LAB isolates with 16S rRNA.

**Figure 2 molecules-26-05874-f002:**
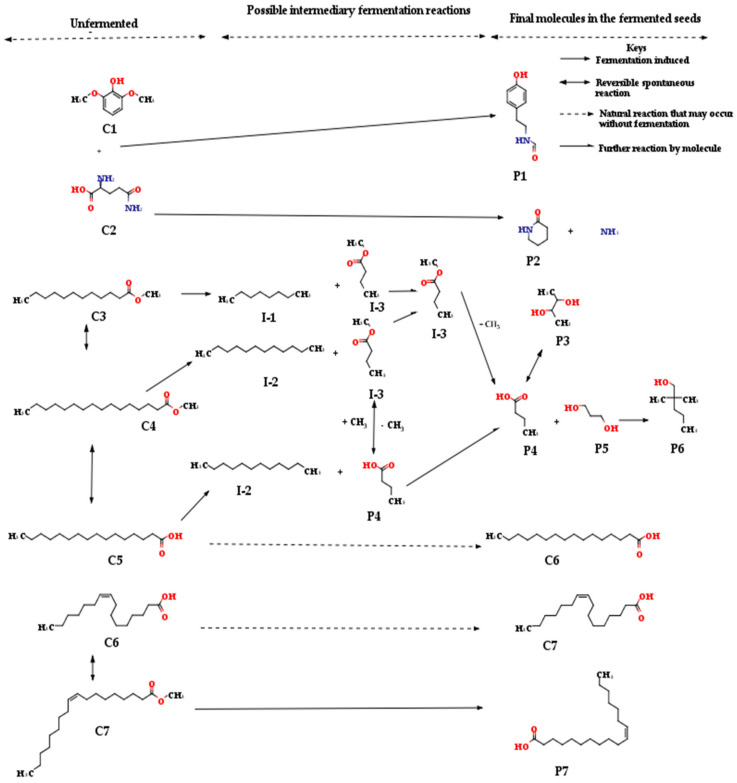
The proposed mechanisms for the transformation of major molecules identified in the natural kernels of *T. catappa* during fermentation. Keys: **P1**: Formamide, N-(p-hydroxyphenethyl) (2.80%); **P2**: 2-Piperidinone (6.63%); **P3**: Butane-2,3-diol (1.81%); **P4**: Butyric acid (16.20%); **P5**: Propane-1,3-diol (19.66%); **P6**: Neoheptanol (2.89%); **P7**: Cis-vaccenic acid (1.69%); **I-1**: Octane; **I-2**: Dodecane; **I-3**: Methyl butanoate; **C1**: Syringol (6.03%); **C2**: Glutamine (1.71%); **C3**: Methyl laurate (1.79%); **C4**: Methyl palmitate (1.53%); **C5**: Palmitic acid (5.20%); **C6**: Palmitoleic acid (2.80%); **C7**: Methyl oleate (2.97%).

**Figure 3 molecules-26-05874-f003:**
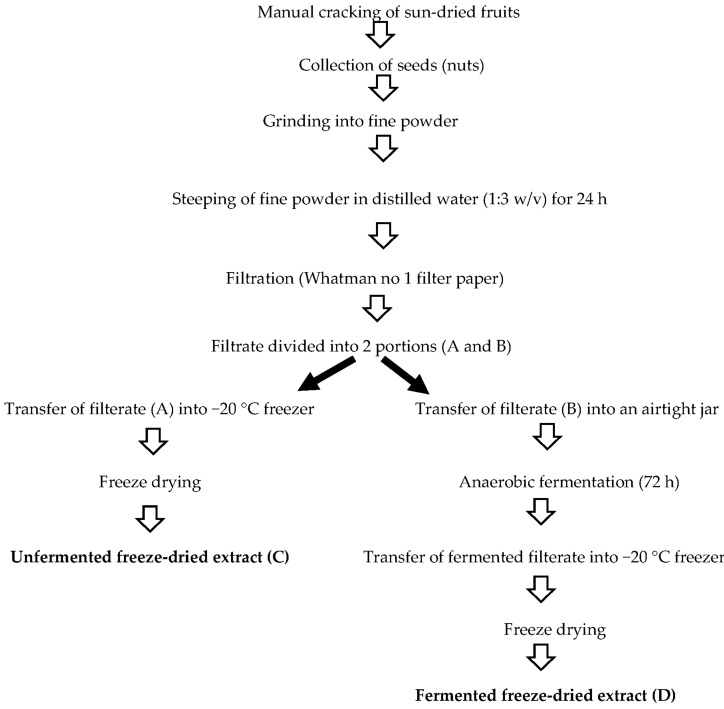
Flow chart illustrating the preparation of flour, unfermented and fermented freeze-dried extracts from *T. catappa* nuts.

**Table 1 molecules-26-05874-t001:** The pH values during fermentation of the seed of *T. catappa*.

Fermentation Treatment	pH
UFTC	7.8
FTC	7.5

Keys UFTC: Unfermented *T. catappa*, FTC: Fermented *T. catappa.*

**Table 2 molecules-26-05874-t002:** Organisms identified using biochemical characterization and genomic identification.

LAB Isolates	Lactic Acid Bacterial	
	Biochemical Characterisation	16sRNA Sequencing (Accession Numbers)
ProbtB1a	*Lactobacillus pentosus*	*Enterococcus faecum* (MW481697)
ProbtB1b	*Lactobacillus casei*	*Enterococcus faecalis* (MW481698)
ProbtB2a	*Lactobacillus casei*	*Enterococcus faecalis* (MW481701)
ProbtB2b	*Lactobacillus casei*	*Enterococcus faecalis* (MW481699)
ProbtB3	*Lactobacillus casei*	*Enterococcus faecalis* (MW481695)
ProbtB4a	Unidentified ^†^	*Enterococcus faecalis* (MW481696)
ProbtB4b	*Lactobacillus plantarum*	Unidentified ^¥^

^†^ Beyond the sensitivity of the biochemical test. ^¥^ Re-identification in-progress.

**Table 3 molecules-26-05874-t003:** Compounds present in the gas chromatography-mass spectrometry (GC-MS) analysis of the unfermented freeze-dried extract of *T. catappa*.

S/N	Peaks	Tr	Area (%)	Similarity Index (%)	Class of Compound	IUPAC Name	Common Name
1	16	11.83	6.03	89	Phenol	Phenol, 2,6-dimethoxy-	Syringol
2	18	12.32	1.71	90	Amino acid	(2S)-2,5-diamino-5-oxopentanoic acid	Glutamine
3	20	13.63	1.79	89	Fatty acid	Methyl dodecanoate	Methyl laurate
4	36	16.87	1.53	90	Fatty acid ester	Methyl hexadecanoate	Methyl palmitate
5	38	17.17	5.20	94	Fatty acid	Hexadecanoic acid	Palmitic acid
6	39	17.74	2.80	85	Fatty acid	Cis-9-Hexadecenoic acid	Palmitoleic acid
7	40	18.03	2.97	91	Fatty acid ester	Methyl (Z)-octadec-9-enoate	Methyl oleate

Keys S/N: Serial number; Tr: Retention time; IUPAC: International Union of Pure and Applied Chemistry.

**Table 4 molecules-26-05874-t004:** Compounds present in the GC-MS analysis of the freeze-dried extract of fermented *T. catappa*.

S/N	Peaks	Tr	Area (%)	Similarity Index (%)	Class of Compound	IUPAC Name	Common Name
1	1	4.51	1.81	91	Alcohol	Butane-2,3-diol	2,3-butanediol
2	3	4.89	16.20	87	Fatty acid	Butanoic acid	Butyric acid
3	4	5.29	19.66	89	Alcohol	Propane-1,3-diol	Trimethylene glycol
4	7	8.35	2.89	85	Alcohol	2,2-dimethylpentan-1-ol	Neoheptanol
5	12	10.06	6.63	96	Piperidones (delta-lactam)	2-Piperidinone	Valerolactam
6	34	17.08	1.18	91	Fatty acid	cis-9-Hexadecenoic acid	Palmitoleic acid
7	35	17.19	4.19	94	Fatty acid	hexadecanoic acid	Palmitic acid
8	37	17.76	1.48	87	Fatty acid	cis-9-Hexadecenoic acid	Palmitoleic acid
9	39	18.17	2.80	86	Amide	Tyramine, N-formyl-	Formamide, N-(p-hydroxyphenethyl)-
10	40	18.35	1.69	89	Fatty acid	(Z)-octadec-11-enoic acid	Cis-vaccenic acid

Keys S/N: Serial number; Tr: Retention time; IUPAC: International Union of Pure and Applied Chemistry.

**Table 5 molecules-26-05874-t005:** The pattern of the sugar metabolism for the isolated lactic acid bacteria (LAB).

S/N	Substrates	Probt B1a	Probt B1b	Probt B4a	Probt B4b
0	Control	-	-	-	-
1	Glycerol	-	+	-	+
2	Erythritol	-	-	-	-
3	D-Arabinose	-	-	-	-
4	L-Arabinose	+	-	-	-
5	D-Ribose	+	+	-	+
6	D-Xylose	+	-	-	-
7	L-Xylose	-	-	-	-
8	D-Adonitol	-	-	-	-
9	Methyl-βD-xylopyranoside	-	-	-	-
10	D-Galactose	+	+	-	+
11	D-Glucose	+	+	-	+
12	D-Fructose	+	+	-	+
13	D-Mannose	+	+	-	+
14	L-Sorbose	-	-	-	-
15	L-Rhamnose	-	-	-	-
16	Dulcitol	-	-	-	-
17	Inositol	-	-	-	-
18	D-Mannitol	+	+	-	+
19	D-Sorbitol	+	+	-	+
20	Methyl-αD-Mannopyranoside	-	-	-	-
21	Methyl-αD-Glucopyranoside	-	-	-	-
22	N-Acetylglucosamine	+	+	-	+
23	Amygdalin	+	+	-	+
24	Arbutin	+	+	-	+
25	Esculin ferric citrate	+	+	-	+
26	Salicin	+	+	-	+
27	D-Cellobiose	+	+	-	+
28	D-Maltose	+	+	-	+
29	D-Lactose	+	-	-	+
30	D-Mellibiose	+	-	-	+
31	D-Saccharose	+	+	-	+
32	D-Trehalose	+	+	-	+
33	Inulin	-	-	-	-
34	D-Melezitose	+	+	-	+
35	D-Raffinose	+	-	-	+
36	Amidon	-	-	-	-
37	Glycogen	-	-	-	-
38	Xylitol	-	-	-	-
39	Gentiobiose	+	+	-	+
40	D-Turanose	-	-	-	-
41	D-Lyxose	-	-	-	-
42	D-Tagatose	+	+	-	+
43	D-Fucose	-	-	-	-
44	L-Fucose	-	-	-	-
45	D-Arabitol	-	-	-	-
46	L-Arabitol	-	-	-	-
47	Potassium gluconate	+	+	-	+
48	Potassium 2-ketogluconate	-	-	-	-
49	Potassium 5-ketogluconate	-	-	-	-

## Data Availability

The accession numbers indicated in the manuscript are for locating the deposited sequences of the identified organisms in the National Centre for Biotechnology Information (NCBI) database—https://www.ncbi.nlm.nih.gov/nucleotide/ (accessed on 3 May 2021).

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
