# Peer review of "Structural Transformation and Creativity Induced by Biological Agents during Fermentation of Edible Nuts from Terminalia catappa"

_molecules, 2021, doi:10.3390/molecules26195874_

Round 1

Reviewer 1 Report

The study is aimed to establish the naturally biotransformed molecules and identify the probiotic agents facilitating the fermentation of Terminalia catappa L. (tropical almond). However, there are some questions must discuss in further.

Major

  1. Authors showed that have four isolates of lactic acid bacteria (LAB) sourced from the fermented T. catappa nuts. However, the bacteria were only analyzed by several biochemical methods, but not genomic sequencing methods, although the Lactobacillus pentosus, Lactobacillus casei, and Lactobacillus plantarum were mentioned in the manuscript, how authors identified them? Moreover, how authors confirmed that these are only lactic acid bacteria (LAB) in the fermentation process?
  2. There so many compounds were not identified by this study.
  3. Authors supposed that palmitic acid and palmitoleic acid exist in both the unfermented and fermented extracts which indicates their functional groups are not modified during this fermentation. However, the palmitic acid (C5 and C6) in Figure 1 can be transformed during the fermentation?

others

  1. In section 2.2, the sentence “This study is probably the first to provide the detailed phytochemical compounds…”, is not a proper statement for this study due to the unconcerned meaning.
  2. The resolution of Figure 1 is too low (unclear).

Author Response

Reviewer 1

  1. Authors showed that have four isolates of lactic acid bacteria (LAB) sourced from the fermented T. catappa nuts. However, the bacteria were only analysed by several biochemical methods, but not genomic sequencing methods, although the Lactobacillus pentosus, Lactobacillus casei, and Lactobacillus plantarum were mentioned in the manuscript, how authors identified them? Moreover, how authors confirmed that these are only lactic acid bacteria (LAB) in the fermentation process?

Response: We have incorporated the details for the genomic sequencing performed during the LAB identification into the manuscript (see page 13, section 4.4.4 –section 4.4.4.2 - methods, page 3, section 2.2 - results, and page 5, section 3.1-discussions) to prove that it was LAB isolated. Authors were quite aware of the possible involvement of other organisms during the fermentation. The lactic acid bacteria (LAB) were the organisms of interest as stated in the objective of the study. That is why a selective media De Man Rogosa and Sharpe agar that allows only for LAB isolation was employed as stated in the manuscript. The API 50CHL kit that was used for the biochemical characterization is also peculiar to identify LAB.

  1. There so many compounds were not identified by this study.

Response: The statement was retained since the many compounds identified in this study are consistent with the findings obtained in other similar plant materials (Xiao and Bay, 2019-EN). Plants, such as the seeds of T. catappa used in this study, are complex in structure with myriads of molecules that may be of health benefits (Basha et al., 2020-EN). This statement had been incorporated into page 8 of the revised manuscript for better comprehension.

  1. Authors supposed that palmitic acid and palmitoleic acid exist in both the unfermented and fermented extracts which indicates their functional groups are not modified during this fermentation. However, the palmitic acid (C5 and C6) in Figure 1 can be transformed during the fermentation?

Response: The statement was changed to 'palmitic acid and palmitoleic acid exist in both the unfermented and fermented extracts, indicating their functional groups are slightly more difficult to modify during this fermentation. It may require a more extended period of fermentation to be completely metabolised. The slight change in acidity (Table 1) during the fermentation of this nut may be responsible for the gradual modification of the acidic functional groups of both compounds.' This statements (Please see page 10, paragraph 2, line 10-13), the corresponding method of determination of the pH (See page 11, section 4.3, line 12-14), and results (Table 1) were also incorporated for more clarity.

others

  1. In section 2.2, the sentence "This study is probably the first to provide the detailed phytochemical compounds…", is not a proper statement for this study due to the unconcerned meaning.

Response: The statement ‘unconcerned meaning’ used by the reviewer is not clear to us. The sentence was therefore retained.

  1. The resolution of Figure 1 is too low (unclear).

Response: Figure 1 (with 800 dpi) was replaced with a higher resolution version (1200 dpi), duplicated molecules (C5 & C6) were merged as one molecule (C5), and re-arrangement of location were performed in new Figure 2 to manage space, and to improve clarity as suggested. The same was also submitted directly uploaded to the journal website for further consideration by the publishing unit.

In general, seven new references (25-29, and 31-32) were added to also increase understanding in the revised manuscript. We appreciate the constructive suggestions that improved the quality of the manuscript.

Reviewer 2 Report

The authors presented interesting manuscript showing the possibility of Terminalia catappa fermentation. Nevertheless, some improvements are necessary. 

  1. The isolated microbial strains should also be confirmed by genetic methods.
  2. The authors mention that seed is rich in vitamin E, however, this compound was not found during fermentation?  Please comment.
  3. It would be advisable to show the quantitative changes of compounds. 
  4. Why was the fermentation carried out in such a large temperature range (26-33 degrees Celsius)? 
  5. Was it checked what other groups of microorganisms appeared during spontaneous fermentation? 
  6. It is necessary to provide the pH value or titrable acidity and microbial counts. 

Author Response

Reviewer 2 - Comments and Suggestions for Authors

The authors presented interesting manuscript showing the possibility of Terminalia catappa fermentation. Nevertheless, some improvements are necessary. 

  1. The isolated microbial strains should also be confirmed by genetic methods.

Response: The detail for the genomic sequencing performed during the LAB identification was incorporated into the manuscript (see page 13, section 4.4.4 –section 4.4.4.2 - methods, page 3, section 2.2 - results, and page 5, section 3.1-discussions) requested.

  1. The authors mention that seed is rich in vitamin E, however, this compound was not found during fermentation?  Please comment.

Response: Analysis of vitamin E is not part of the objectives, and the indicated statement on vitamin E was not from this experimental study. It was a finding from another study that adopted to introduce the article (appropriate reference was provided). Also, the study is limited to the phytochemicals that could be detected by the GC-MS used in this study. The vitamin E was not detected by the method used for analysis.

  1. It would be advisable to show the quantitative changes of compounds. 

Response: No response, since Table 3 and Table 4 in the revised manuscript, and the supplementary table (S1 and S2) has already presented both the quality (individual molecules involved) and their corresponding quantity (amount) of the identified compounds. These were further substantiated with a proposed mechanisms of action for the changes of the compounds.

  1. Why was the fermentation carried out in such a large temperature range (26-33 degrees Celsius)? 

Response: The temperature range was as recorded for the natural environment used to ferment the seed. The uncontrolled temperature is typical of all-natural fermentation types. Natural fermentation can depict the types of organisms and molecules generated during processing of most food for household consumption. The statement was already confirmed in the introduction of the manuscript (page 2).

  1. Was it checked what other groups of microorganisms appeared during spontaneous fermentation? 

Response: The other groups of microorganisms were not considered since the study's objective was focused on the health-beneficial lactic acid bacteria (LAB) associated with the fermentation. A selective media (de Man Rogosa Sharpe, MRS) agar that encourage the growth of LAB, but exclude the growth of other microbes was therefore used to isolate the organism during the experimentation. This statement was incorporated into the materials and methods of the manuscript to improve understanding (See page 11, section 4.4.1, line 4-5).

  1. It is necessary to provide the pH value or titrable acidity and microbial counts. 

Response: The method for determining the pH (See page 11, section 4.3, line 12-14) and the results (Table 1) obtained were incorporated as suggested. Also, a more detail information than the requested microbial counts (including the genomic sequencing results) on the organisms was incorporated into the revised manuscript.

In general, seven new references (25-29, and 31-32) were added to also increase understanding in the revised manuscript. We appreciate the constructive suggestions that improved the quality of the manuscript.

Round 2

Reviewer 1 Report

  1. In table 1, the acidity is only slightly changed (0.3), how to concluded that it responsible for the gradual modification of the acidic functional groups of both compounds?
  2. In section 2.2, the sentence "This study is "probably" the first to provide the detailed phytochemical compounds…", which is not a proper statement for this study due to the "uncertain" meaning.

Author Response

Dear Reviewer,

We have attended to the two comments raised. The comments were indeed beneficial, constructive and have improved the quality and understanding of the manuscript. Authors appreciate the efficient and fast review process. Finally, twelve new references (58 - 69) were added to support the justification added to the revised manuscript. Kindly find below our point-by-point response to each of the comments by the reviewers.

Yours faithfully,

Israel S. Afolabi

Reviewer 1

  1. In table 1, the acidity is only slightly changed (0.3), how to concluded that it responsible for the gradual modification of the acidic functional groups of both compounds?

Response: The modification of the acidic functional group by the slight change in pH is possible. We have incorporated more statements (see page 10, paragraph 1, line 11-13 and line 15-26) into the manuscript to increase understanding. The statements are as indicated below: -

The two compounds were not completely metabolised like other compounds completely eliminated after the fermentation indicates a gradual modification process. An extended period of fermentation beyond the 72 h used in this study may completely metabolise these compounds (palmitic acid and palmitoleic acid). This fermentation of T. catappa produces a pH change of 3.0, similar to the slight pH change earlier reported during fermentation [58]. The slight difference in acidity was sufficient to induce a change in the activity of microorganisms that drives the modification of compounds [58,59]. A pH change from 6.32 to 4.25 (a margin of 2.07) produced conspicuous biochemical changes during fermentation [60,61]. Other similar fermentation studies with pH change from 5.0 to 7.00 (a margin of 2.0) resulting in a chemical modification depend on the type of compounds present during the fermentation. The pH change did not affect the amino acids but immensely impacted the carbohydrate structures [62]. Such slight change in acidity led to the modification of polyphenols, γ-aminobutyric acid, citric acid, and other compounds during microbial fermentation or in other living systems [63-69]. These findings also signify that T. catappa nut is a rich source of dietary fatty acid in raw and fermented form.

2. In section 2.2, the sentence "This study is "probably" the first to provide the detailed phytochemical compounds…", which is not a proper statement for this study due to the "uncertain" meaning.

Response: The word “probably” was deleted from the sentence to make it more certain (see section 2.3, line 1).

Reviewer 2 Report

I reanalyzed the document and the answers given by the authors, and consider that the document was clearly improved, thus it can be accepted.

Author Response

Dear Reviewer,

Authors appreciate the efficient and fast review process that has immensely improved quality of the manuscript. The acceptance is also appreciated.

Regards.